# Anguillid Eels as a Model Species for Understanding Endocrinological Influences on the Onset of Spawning Migration of Fishes

**DOI:** 10.3390/biology11060934

**Published:** 2022-06-19

**Authors:** Ryusuke Sudo, Takashi Yada

**Affiliations:** 1Fisheries Technology Institute, Minamiizu Field Station, Japan Fisheries Research and Education Agency, Minamiizu, Kamo, Shizuoka 415-0156, Japan; 2Fisheries Technology Institute, Nikko Field Station, Japan Fisheries Research and Education Agency, Chugushi, Nikko 321-1661, Japan; yadat@affrc.go.jp

**Keywords:** anguillid eels, silvering, spawning migration onset, 11-ketotestosterone

## Abstract

**Simple Summary:**

Endocrine regulation has been thought to play a major role in the onset of migration. Anguillid eels provide a good model for studying the onset mechanisms of migrations to breeding areas, because the process of the onset of migration occurs in inland waters. In this review, we summarize information about the silvering process in anguillid eels and the dynamics of mRNA expression of neurohormones and pituitary hormones, thyroid hormones, and sex steroids associated with the onset of the spawning migration. We also provide new results. Because 11-KT drastically increases during silvering, the role of 11-KT in the onset of spawning migration was discussed in detail.

**Abstract:**

Anguillid eels are the iconic example of catadromous fishes, because of their long-distance offshore spawning migrations. They are also a good model for research on the onset mechanisms of migrations to breeding areas, because the migrations begin in inland waters. When eels transform from yellow eels to silver eels, it is called silvering. Silver eels show various synchronous external and internal changes during silvering, that include coloration changes, eye-size increases, and gonadal development, which appear to be pre-adaptations to the oceanic environment and for reproductive maturation. A strong gonadotropic axis activation occurs during silvering, whereas somatotropic and thyrotropic axes are not activated. Among various hormones, 11-ketotestosterone (11-KT) drastically increases during spawning migration onset. Gradual water temperature decreases simulating the autumn migratory season, inducing 11-KT increases. Administration of 11-KT appeared to cause changes related to silvering, such as early-stage oocyte growth and eye enlargement. Moreover, 11-KT may be an endogenous factor that elevates the migratory drive needed for the spawning migration onset. These findings suggested that water temperature decreases cause 11-KT to increase in autumn and this induces silvering and increases migratory drive. In addition, we newly report that 11-KT is associated with a corticotropin-releasing hormone that influences migratory behavior of salmonids. This evidence that 11-KT might be among the most important factors in the spawning migration onset of anguillid eels can help provide useful knowledge for understanding endocrinological mechanisms of the initiation of spawning migrations.

## 1. Introduction

Migratory fishes are one of the most fascinating types of fishes because they can swim across long distances through different aquatic habitats for feeding and growth or reproduction. However, the mechanisms for how they begin their remarkable migrations have often been hard to determine. Endocrine regulation has been found to play an important role in triggering the onset of migration [1]. Anadromous salmonids make downstream migrations from rivers into the ocean (or in landlocked lakes), after a transformation from the parr to the smolt stage. During the smoltification period and downstream migration, various hormones including cortisol, growth hormone, prolactin, thyroxine (T4), and triiodothyronine (T3), are increased in smolts [2,3,4,5]. Among these hormones, T4 is recognized as one of the most important endocrine factors involved in salmonid smoltification and downstream migration, since T4 sharply increases (T4 surge) during the downstream migration in several salmonid species [6,7,8]. T4 is also an important factor in the upstream migration of amphidromous ayu, *Plecoglossus altivelis*, which spawn in freshwater, with the larvae drifting downstream to the coastal ocean for an extended larval growth period before they re-enter freshwater for juvenile growth until maturity [1,9]. These migrations are directed to the growth area from the breeding area. However, there is little known about the endocrinological mechanisms of the onset of migration directed to the breeding areas from growth areas.

The catadromous anguillid eels have a uniquely complex type of life history and are one of the most famous migratory fish species (Figure 1), because their adult eels were found to migrate thousands of kilometers to offshore spawning areas. Their leaf-like transparent leptocephalus larvae are transported by ocean currents from their spawning area toward coastal areas, where metamorphosis into the glass eel stage occurs and their pelagic behavior changes to being demersal and they begin inshore migration. They then settle in brackish water estuaries or freshwater rivers for the yellow eel juvenile growth stage. After their multi-year juvenile growth period, eels transform from yellow eels to the migratory-stage adult eels through the silvering process, and then the silver eels start their initial spawning migration out of freshwater and coastal waters and then through the offshore ocean. Therefore, the silvering process of anguillid eels provides a good model for research on the mechanisms of the onset of migrations directed to breeding areas, because the processes of spawning migration onset occur in inland waters where it is easier to observe than in the sea.

In this review, we firstly summarize the characteristics of silvering, which is commonly used as an indicator for the initiation of the anguillid eel spawning migration. Then, the endocrinological influences on the onset of the spawning migration are reviewed by describing the changes in various hormones (pituitary hormones, thyroid hormones, and sex steroids) associated with silvering and the spawning migration. In addition, we newly reported the changes in neurohormones during silvering and the relationship between androgen and neurohormone.

## 2. Silvering

One of the most obvious modifications to yellow eels that occurs during the process of silvering are changes in body coloration. Growth-stage yellow eels generally appear greenish dorsally and yellow-white ventrally. During the beginning of the spawning migration, they transform into silver eels that have a black colored dorsum and a silvery, light-color lower body. This body color modification is why this process is called silvering, which is typically used for stage-determination of reproductive migration-phase anguillid eels, and the pectoral fins also become black [10,11,12,13].

In addition to body-color modifications, various morphological changes are observed during silvering. One obvious change is that relative eye-size increases during the silvering of anguillid eels. In European eels, this index is one of the characteristics used to distinguish migrating silver eels from the non-migrating yellow eels [14]. Clear swim bladder modifications also occur in the eels during silvering when the rete mirabile becomes enlarged in silver eels, and more crystalline-guanine deposition occurs in the swim bladder of silver eels also increases [15,16,17]. It is also well-known that degenerative morphological change occurs in the stomach and intestines of silver eels [13,18,19], and silver eels were indicated to not feed during their oceanic migrations [20]. All these changes, including body-color modification, seem related to adapting to the conditions of the oceanic environment.

The onset of gonadal maturation is another characteristic of silvering. The gonadosomatic index (GSI, the % gonad weight of body weight) of female eels increases progressively during the silvering process. Gonad development is directly correlated to the other morphological changes that occur during silvering, such as skin color modifications and eye-size enlargement [10,11,12,13,21,22]. The GSI increase can be used as a good criterion for estimating the advancement stage of the European eel silvering process [12]. 

## 3. Neurohormones

Within the brain, several neuropeptides and amines are important modulators for the migratory behavior in some vertebrates. Gonadotropin-releasing hormone (Gnrh) is the stimulating factor of the hypothalamus-pituitary-gonadal axis, which is the central pathway for the mediation of sexual maturation and reproduction in the endocrine system. In fish, Gnrh appears to play a role in the regulation not only of maturation, but also of behavior, including the homing migration of anadromous salmon [23]. The regulation of corticosteroid secretion in relation to the stress responses and glucose metabolisms has been established as the hypothalamus-pituitary-interrenal axis in fish [24,25]. In salmonids, the intracerebroventricular administration of corticotrophin-releasing hormone (Crh) results in a stimulation of downstream movement, or changes in aggression and anxiety-like behaviors [26,27,28].

Possible contributions of the neurohypophysial peptides, arginine–vasotocin, and isotocin were shown to influence the regulation of fish social behavior, especially in the aggressiveness for territory formation [29,30,31,32,33]. Brain monoamines, such as dopamine, serotonin, and gamma-aminobutyric acid (Gaba) are well known as neurotransmitters related to emotional responses in mammals, and are used as psychosomatic medicines [34,35]. The effects of those monoamines on behavior have also been shown in fish species [36,37,38,39]. However, the involvement of the neurohypophysial peptides and brain monoamines in fish migratory behavior is not known well, especially in silvering eels.

In eels, two types of Gnrh occur which are the mammalian Gnrh (mGnrh) and the chicken Gnrh-II (cGnrh-II) [40,41]. Immunocytochemical localization analysis showed that mGnrh-immunoreactive fibers were present in many brain areas and in the pituitary of eels [40]. We measured mRNA expression of *mgnrh*, and tyrosine hydoroxylase (*th*), which is the rate-limiting enzyme in the biosynthesis of dopamine, in yellow and silver eels, which were collected from a brackish lake (Figure 2), for the first time. mRNA expression levels of *mgnrh* and *th* in the olfactory bulb and hypothalamus was measured using quantitative real time PCR (qPCR). *mgnrh* expression levels in the hypothalamus were significantly higher in the silver eels than yellow eels, whereas no differences occurred in the olfactory bulb. The expression of *th* was higher in the olfactory bulb in silver eels, and there was no significant difference between silver and yellow eels in the hypothalamus. This suggests that these neuro hormones are involved in the onset mechanism of the spawning migration of eels. 

## 4. Growth Hormone, Prolactin, Somatolactin

In teleost fishes, growth hormone (Gh), prolactin (Prl), and somatolactin (Sl) function to control pleiotropic biological functions and have originated from an ancestrally common molecule [43]. Gh has been suggested to be involved in the regulation of somatic growth in teleosts [44,45,46]. Gh also functions for seawater adaptation as a hypo-osmoregulatory hormone in fishes [47]. However, teleost Prl seems to function as a hyper-osmoregulatory hormone involved in freshwater adaptation [47,48]. Sl functions in teleosts for energy mobilization, stress response, metabolism of calcium, acidosis, and pigmentation [49], although there is little information about its osmoregulatory functions. The Gh/Prl/Sl group of hormones also was implicated in reproduction in some fishes [43,50,51,52,53,54,55,56]. Furthermore, Gh and Prl function in the regulation of the downstream migration of salmonids [2,5,57].

Our groups examined the pituitary mRNA expression of these various hormones in Japanese eels in relation to their migratory season, different salinity environments, silvering, and downstream migration or not as described below (Figure 3) [58]. Female Japanese eels were caught in a brackish lake and freshwater river inlets during July–December. The silver eels were present during October–December in both the lake and river, and thus the period from July to September was defined as the non-migratory season, and the period from October to December as the migratory season. The habitat-use histories of these eels were determined using Sr:Ca otolith microchemistry techniques (see Sudo et al. 2013 for detailed methods) [58]. Expression levels were determined by qPCR.

This research on the Japanese eel showed that *gh* mRNA expression increased in eels in the freshwater rivers from non-migratory season to migratory season, and clearly decreased during silvering (Figure 3), which is similar to other migratory species. In European eels, *gh* mRNA expression and Gh plasma levels were not activated during silvering [59,60]. Gh has an important role in anadromous salmonids during smoltification along with thyroid hormones, and large increases of Gh have been reported during smoltification [4,61]. Gh may affect various types of behavior, including salinity preference, feeding and predator avoidance, which are migratory-related activities [62,63,64]. On the contrary, Gh may have no, or an inhibitory, role for silvering and/or the spawning-migration onset in catadromous eels.

It is well known that Prl is involved in activating migratory motivation in some bird families [65]. Prl is also related to the regulation of migration in ayu [66]. During silvering, *prl* mRNA expression was unchanged in the eels from the brackish lake (Figure 3). Although there no significant statistical differences were found between yellow and silver eels, *prl* mRNA expression decreased in eels in the freshwater river. A previous study reported that Japanese eel *prl* mRNA levels decreased significantly during silvering [67]. Water intake, which is crucial for avoiding dehydration in seawater is inhibited by injection of Prl [68]. The decrease of *prl* mRNA expression in eels during silvering in the freshwater river may be related to preparation for adaptation to the marine environment.

Similar to *prl* mRNA expression, there was a significant difference in *sl* mRNA expression between yellow eels and silver eels in the river, whereas there was no significant difference in the brackish lake. In addition, *sl* mRNA expression was significantly decreased after downstream migration. In contrast to eels, *sl* mRNA expression showed increases in chum salmon, *Oncorhynchus keta*, before the start of their homing behavior [69]. Although it is still unclear why Sl changes in diadromous fishes, this difference may be a reflection of whether the migratory pattern is anadromous or catadromous.

## 5. Thyroid Hormones and Thyroid-Stimulating Hormone

The thyroid hormones, thyroxine (T4) and triiodothyronine (T3) are critical for the development, growth, and metabolism of vertebrates, and are regulated by the pituitary glycoprotein thyroid-stimulating hormone (Tsh). Thyroid hormones are well known to be involved in the metamorphosis of amphibians [70], and various fishes, including the Japanese flounder, *Paralichthys olivaceus* [71]. Thyroid hormones also have been linked to the migratory behavior of some types of fishes. In Atlantic salmon, *Salmo salar*, for example, juveniles exhibited a strong increase in blood T4 just before downstream migration [72]. It also has been shown that T4 was important for the initiation of upstream migration in juvenile ayu [73]. Similarly, it also seems that thyroid hormones are important for controlling glass eel upstream migration [74,75].

We previously showed that no significant differences occurred in the thyroid hormone levels in the plasma between silver and yellow eels (Figure 4). This is in congruence with other research findings showing moderate increases in T4 and a lack of significant variations in T3 during silvering [59,76,77]. In addition, T3 treatment did not cause eye-size or digestive tract changes in yellow eels [64]. Our group and other studies also measured mRNA expression of the transcription of the β subunit of Tsh (*tshβ*). A significant increase of *tshβ* was observed in eels in the brackish lake during silvering, while there was no significant difference between silver and yellow eels in freshwater rivers (Figure 3). In other studies, measurement of *tshβ* mRNA expression showed weak, non-significant increases in expression between yellow and silver eels [59,77]. This indicated that the thyrotropic axis is not or is very weakly linked to eel silvering and the onset of spawning migration.

## 6. Gonadotropins

Silvering marks the onset of gonadal development in anguillid eels, so gonad weight increases during silvering, and reproductive hormone changes also occur. Histological observations indicated that the oocytes of silver eels were mostly at the primary yolk globule stage, but the oocytes of yellow eels were mostly at the early oil drop stages of Japanese eels [78]. In vertebrates in general, gonadal development is regulated by gonadotropins that consist of follicle-stimulating hormone (Fsh) and luteinizing hormone (Lh) [79]. The regulation of gonadal development by gonadotropin was also confirmed in eels, using recombinant gonadotropins [80,81]. Gonadotropins also are involved in controlling sex steroids, which are related to the migratory behavior of masu salmon, *Oncorhynchus masou* [82].

Both transcription of Fsh β subunit (*fshβ*) and Lh β subunit (*lhβ*) significantly increased during silvering, although the statistical difference was not observed in river eels (Figure 3) [42]. After downstream migration, there was a tendency for *fshβ* to decrease and *lhβ* to increase. Han et al. [83] also showed that *fshβ* and *lhβ* were increased during silvering in Japanese eels. It has also been reported that migrating tropical Celebes eels (*Anguilla celebesensis*) exhibited significantly higher levels of *fshβ* and *lhβ* mRNA expression than non-migrating eels [84]. In European eels, *fshβ* mRNA levels increased during the early silvering stages, which was followed by later increases in *lhβ* mRNA levels [59]. Similar expression patterns of gonadotropins were observed in New Zealand short-finned eels, *Anguilla australis* [85]. These studies suggest that both Fsh and Lh are involved in silvering, with Fsh acting at the early stage of the silvering process and Lh at a later stage.

## 7. Sex Steroids

Sex steroids produced in the gonads are regulated by gonadotropins. Among these sex steroids, estrogen (estradiol-17β, E2) functions for vitellogenesis in the ovaries, and androgens (testosterone, T; 11-ketotestosterone, 11-KT) regulate spermatogenesis in the testis. Along with their roles in fish reproduction, sex steroids contribute to fish growth [86], body composition changes [87], intermediary metabolism [88], osmoregulation [89] and migration [90].

In female eels, the plasma E2 level of silver eels is slightly, but significantly higher than that of yellow eels, both in freshwater rivers and the brackish lake (Figure 4). Increases in the E2 of females during silvering were also confirmed in New Zealand long-finned eels, *A. diffenbachii*, New Zealand short-finned eels [91,92], Japanese eels [93], and European eels [59]. In migrating Celebes eels, plasma E2 levels were significantly higher than in non-migrating eels [84]. However, there was no increase in E2 after downstream migration. In addition, male eel plasma levels of E2 did not change during the silvering process [42,93]. E2 seems to be involved in the maturation of female eels, but not in their migratory behavior.

In contrast with E2, T increases have occurred when silvering occurs in both male and female eels of several species [42,58,91,92,93]. 11-KT also increased during silvering for both male and female New Zealand freshwater eels and in the Japanese eel [84,91,92], even though 11-KT has traditionally been viewed to be a male fish hormone. In addition, 11-KT drastically increased after downstream migration (Figure 4). Increases in the levels of 11-KT after downstream migration were also reported in Celebes eels [84]. These results suggest that androgens particularly 11-KT could play a major role in the onset of spawning migration in anguillid eels.

## 8. Role of Androgen in the Onset of Spawning Migration

Measurements of hormones during silvering have shown that the gonadotropic axis clearly becomes activated. Among reproductive hormones, androgens have been shown to be important in eel silvering. It was shown that 17α-methyltestosterone injections into male silver European eels resulted in eye diameter enlargement, increased skin thickness, and head and fin darkening [94]. Similarly, T-implants induced eye size increases in male silver-stage European eels [95]. Aroua et al. (2005) also found that treatment with T-induced gut-index decreases and an in eye-index increases, but E2 had no effect in the European eel [58]. In the New Zealand short-finned eel, eye-diameter increases, skin-thickening, and other silvering-related changes were stimulated by 6-weeks of 11-KT implants, which is a non-aromatizable androgen [96]. In addition, our research found that 11-KT treatment caused the growth of early-stage oocytes, eye-size enlargement, digestive tract degeneration, and swim bladder development for Japanese eels [84]. These findings strongly suggest that androgen could induce silvering.

Decreases in water temperatures are an important factor to stimulate silvering, which then causes the eels to begin their spawning migrations. Silver eels of all temperate eels are mostly caught during autumn and winter when temperatures are decreasing (European eel: Vøllestad et al. 1986, Durif and Elie 2008; American eel: Haro 1991, Japanese eel: Okamura et al. 2002, Sudo et al. 2017; New Zealand short-finned and New Zealand long-finned: Todd 1981) [19,97,98,99,100,101]. Durif and Elie (2008) also showed that spawning migration onset could be affected by unusually low water temperatures even during the summer [98], which induced earlier migration timing. A similar finding was also reported in Japanese eels [19]. Thus, we applied gradual water temperature decreases (25–15 °C) to captive eels that simulated the autumn migration season temperature changes and examined the effect on plasma levels of androgens [102]. Plasma levels of 11-KT increased with water temperature decrease, while T did not. This result suggests that 11-KT increases when water temperatures decrease in autumn and induces eel silvering.

It has been reported that 11-KT not only promotes silvering in eels, but also affects the motivation for migration. In migratory birds, they express high locomotor activity during the migratory season which is called migratory restlessness (also known as Zugunruhe), and this elevated activity appears to be an indicator of the urge to migrate [103]. Our study showed that Japanese eel silver eels had higher locomotor activity and reduced negative phototaxis behavior during the spawning migration season in comparison to yellow eels [104]. This type of stage-specific increase in locomotor activity that occurred in enclosed aquaria appears to be an expression of migratory restlessness, because the eels were shielded from meteorological factors. Therefore, their migratory restlessness behavior may reflect the internal motivation of eels to start their spawning migrations in the same way as migratory restlessness is expressed in birds. In addition, 11-KT administration induced higher levels of activity in yellow eels that do not migrate, which suggests that hormones may be directly involved in elevating the drive for the spawning migration in silver eels. In New Zealand short-finned eels, 11-KT treatment caused higher frequencies of movements between fresh water and seawater, which is also likely to be related to migratory restlessness [105]. In addition, in European eels, androgen stimulated brain dopaminergic systems that may influence specific types of behavior [106]. This showed that 11-KT is not only involved in silvering, but also in the behavioral onset of the spawning migration. All these findings suggested that water temperature decreases in autumn resulted in 11-KT to be increased, and this would stimulate silvering and elevated migratory drive.

This directly contrasts with the downstream migration patterns of salmonid fishes. For example, precocious male masu salmon that were found to have relatively high plasma androgen levels, did not exhibit downstream migration behavior [107,108,109]. Downstream migration of Atlantic salmon was also inhibited by androgen administration [110], and this also occurred in masu salmon [111]. This is to be expected it seems, because any effect of androgens on downstream migration should be clearly different in the two types of diadromous fishes due to their opposite-direction migratory patterns. In adult catadromous anguillid eels, androgens increased during their downstream migration, but in anadromous juvenile masu salmon they inhibit migration [111]. This is likely related to the quite different types of life history stages that occur in salmon and eels, and the motivational differences between the eels that migrate for spawning, and the salmon that migrate for feeding and growth.

## 9. Interaction of Hormones

It seems likely that the combination of field research and laboratory experiments can be effective to find and examine possible hormonal interactions when the spawning migrations of anguillid eels are beginning. Among endocrine factors, which seem to be candidates for being key players during the spawning migration onset of Japanese eels as described above, 11-KT seems useful to examine further. We collected new data that showed the relationship between plasma levels of 11-KT and *crh* mRNA levels detected in the preoptic area of the hypothalamus in female Japanese eels captured in riverine, lacustrine, and estuarine areas around Japan, which had body lengths longer than 30 cm. There was a significant positive correlation (Spearman test, r = 0.325, *p* < 0.001), suggesting a possible influence of elevated 11-KT levels on *crh* expression in the central nervous system (Figure 5). To evaluate this hypothesis, we firstly examined the effects of 11-KT administration through an osmotic minipump (alzet, CA) inserted into the body cavity of eels (~30 cm length) in a laboratory condition for three days. Figure 6 shows that the intraperitoneal insertion of a minipump could release 11-KT and create levels that were comparable to maturating silver eels. We also newly showed that *crh* mRNA contents of the preoptic area were significantly elevated by the administrated 11-KT (Mann-Whitney U test, *p* < 0.05), which suggests that the increased circulating levels of 11-KT that occur during silvering, might induce the onset of spawning migratory behavior of eels through interaction with *crh* expressed within the brain with associated factors in the HPI axis, including corticosteroids.

## 10. Conclusions

Catadromous anguillid eels are unique among all fishes because they undergo a silvering process in freshwater that accompanies early gonadal maturation, which includes morphological changes to their eyes, swim bladder, and body coloration in preparation for their oceanic spawning migrations to offshore waters. Various hormones change during the onset of their spawning migrations, among which 11-KT drastically increases. Gradual water temperature decreases appear to simulate the autumn migratory season and 11-KT elevation. Administration of 11-KT appears to induce silvering-related changes such as early oocyte growth, eye enlargement, digestive tract degeneration, and swim bladder development. Moreover, 11-KT may be an endogenous factor that elevates the migratory drive that is needed to start the spawning migration. Thus, it appears likely that 11-KT is one of the most important factors in the onset of the spawning migration in eels. Although there may be an association between 11-KT and Crh, based on data published here, that might be linked to influencing the motivation of migratory behavior, the endocrinological regulation mechanism of 11-KT is unclear. Future studies should examine the endocrinological mechanisms of 11-KT elevation and the effect of 11-KT on neurohormones.

## Figures and Tables

**Figure 1 biology-11-00934-f001:**
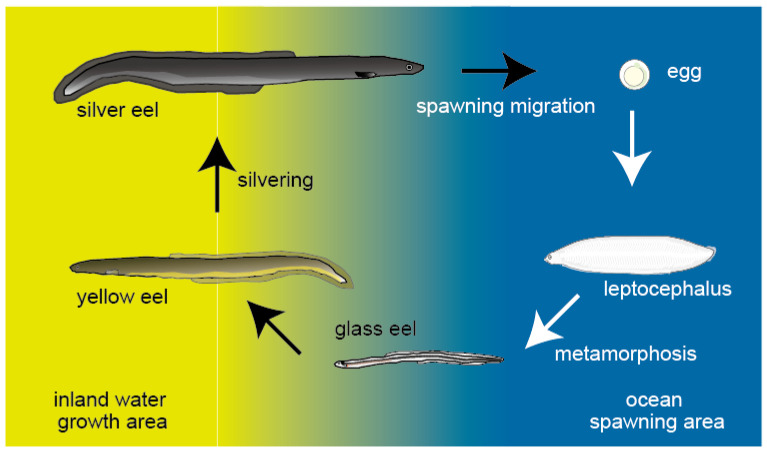
The anguillid eel catadromous life history with spawning and larval development occurring in the offshore ocean and juvenile growth occurring in freshwater or estuarine habitats.

**Figure 2 biology-11-00934-f002:**
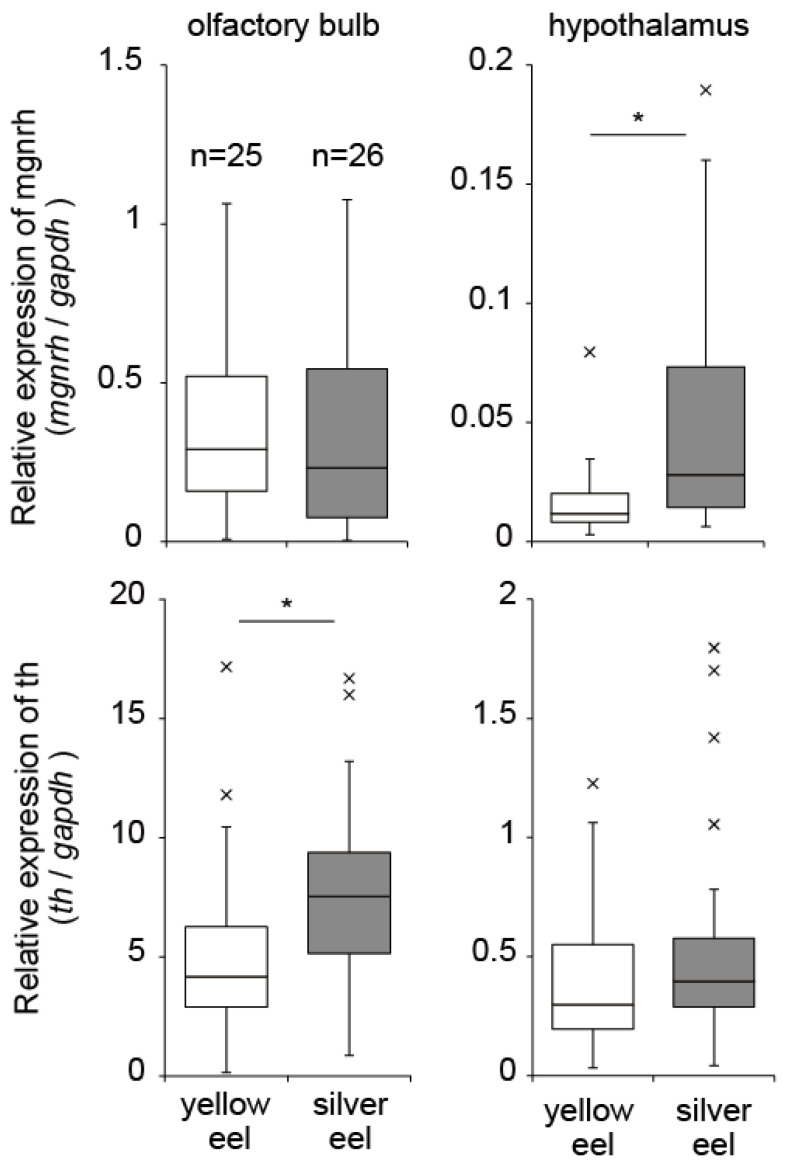
Relative expression levels of *mgnrh* and *th* in the olfactory bulbs and hypothalamus of Japanese eels (*A. japonica*) during silvering, as quantified by quantitative real time PCR. Asterisks indicate significant differences between yellow eels and silver eels (*p* < 0.05; U-test). Cross marks indicate outliers. All samples were collected from a brackish lake (Hamana lake, Shizuoka prefecture, Japan). Eels were classed as yellow or silver, according to a silvering index [11]. Eels (*n* = 51) were sacrificed by decapitation while still anesthetized, and then their brains were quickly removed for molecular biological analysis. Expression levels of target genes (*mgnrh* and *th*) and a reference gene (glyceraldehyde-3-phosphate dehydrogenase, *gapdh*) were measured using qPCR. Primers (mgnrhfw: 5′-GACACCTCCAGTTTGCCCA-3′; mgnrhrv: 5′-TTGCCAGTATTTCCTTCAGGCT-3′; thfw: 5′-CTGCGTTCACGAGCTCTTAGG-3′; thrv: 5′-CAAGGCCAATGTTCTGAGAAAAC-3′) and these assays were designed by Primer Express Software. For more detailed methods on the measurement of mRNA expression by qPCR, see Sudo et al. [42].

**Figure 3 biology-11-00934-f003:**
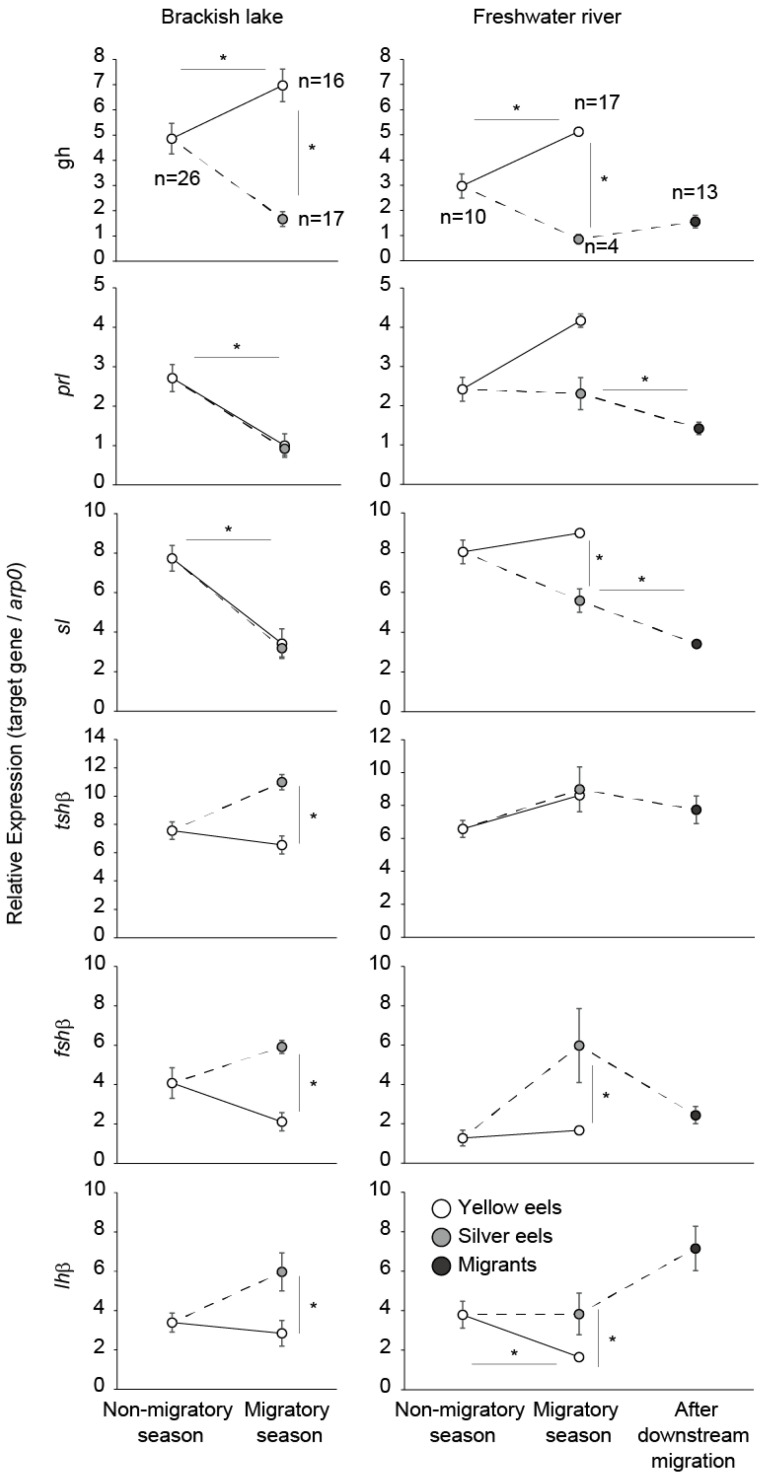
Seasonal differences of mRNA-expression of pituitary hormones for female yellow eels and silver eels (*A. japonica*) from a brackish lake and its freshwater inlet rivers. White circles indicate yellow eels, gray circles show silver eels, and black circles show migrants, which are the silver eels after downstream migration. Data are means ± standard error. Asterisks show significant differences between each group (*p* < 0.05; Steel-Dwass). This figure is based on data from Sudo et al. 2011 [42] and Sudo et al. 2013 [58].

**Figure 4 biology-11-00934-f004:**
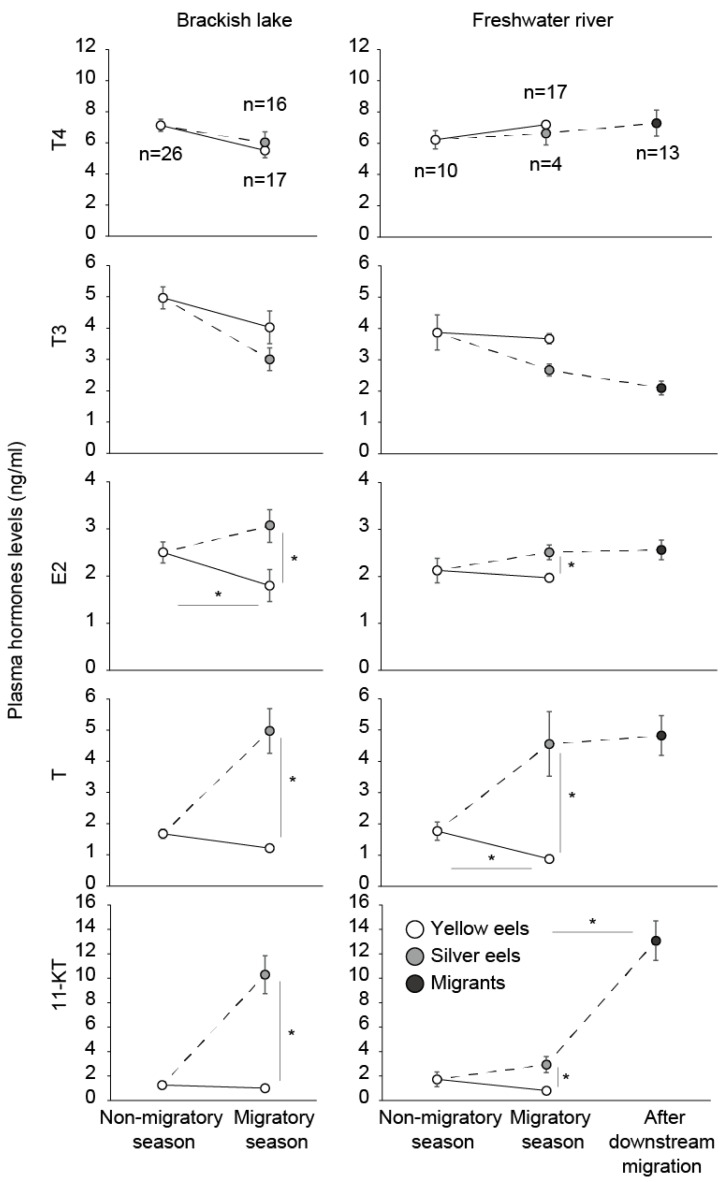
Seasonal changes of the steroid hormones (E2, T and 11-KT) and the thyroid hormones (T4 and T3) for each type of female Japanese eels (*A. japonica*) from both the brackish lake and its freshwater inlets. This figure is based on data from Sudo et al. 2011. Asterisks show significant differences between each group (*p* < 0.05; Steel-Dwass).

**Figure 5 biology-11-00934-f005:**
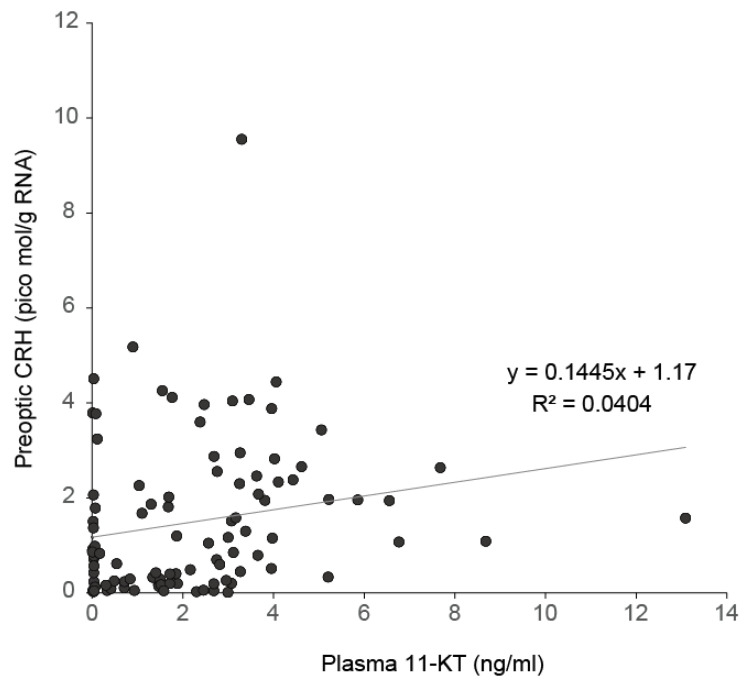
Correlation between plasma 11-KT and preoptic *crh* mRNA levels in wild female Japanese eels (*A. japonica*) (n = 100). Blood plasma was extracted with diethyl ether as following de Jesus et al. [112]. Analyses were made in duplicate on each sample using the 11-Keto-Testosterone ELISA Kit (Item No. 582751, Cayman Chemical Inc., ANN Arbor, MI, USA) according to the manufacturer’s instructions. Brains of each fish were dissected out under microscopy, mixed with ISOGEN (Nippon Gene), and total RNA was isolated according to the manufacturer instructions. Total RNA was treated with RNase-free DNase I (Takara). After inactivation of DNase, reverse transcription was carried out using the SuperScript II First-Strand Synthesis System (Invitrogen). QPCR for crh was performed with LightCycler 480 Instrument (Roche Diagnostics) as described by in Yada et al. [113]. The sequences of primers in the assays to amplify crh were 5′-TCACGCAGCGTCTTTTGC-3′, forward; 5′-GCTGGCTAGCGTAGCTGCTT-3′, reverse, which were designed based on the Gene Bank accession number LC01940.

**Figure 6 biology-11-00934-f006:**
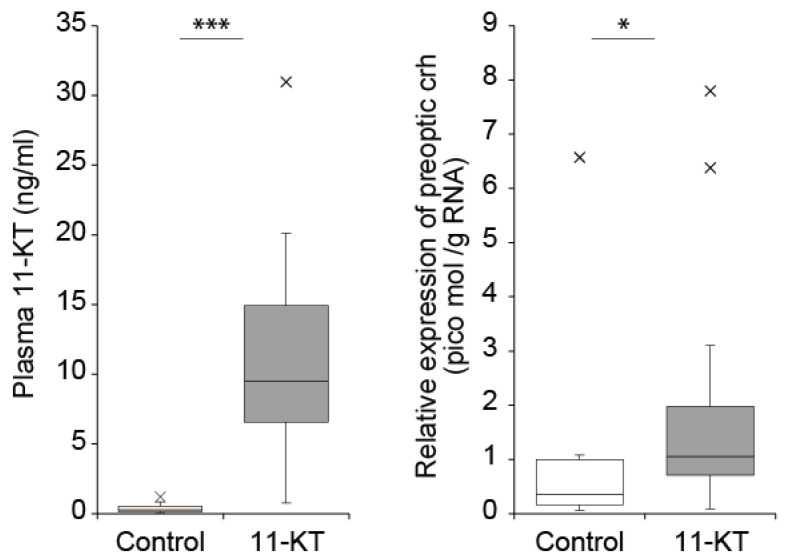
Plasma 11-KT and preoptic *crh* mRNA levels in female Japanese eels (*n* = 12) (*A. japonica*) with intraperitoneal administration of 11-KT by osmotic minipump. Data are expressed as Tukey’s boxplots. *, *** significantly different from the control group at *p* < 0.05, 0.001 by Mann-Whitney U test, respectively. Cross marks indicate outliers.

## Data Availability

The data in this study are available upon request from the first author.

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
