# Peer review of "Anguillid Eels as a Model Species for Understanding Endocrinological Influences on the Onset of Spawning Migration of Fishes"

_biology, 2022, doi:10.3390/biology11060934_

Round 1

Reviewer 1 Report

The authors presented an interesting knowledge about the silvering process in anguillid eels and the dynamics of mRNA expression of neuro hormones and pituitary hormones, thyroid hormones and sex steroids associated with the onset of their spawning migration. This manuscript has a merit to be published in Biology.

There are some suggestions which would improve the quality of this manuscript.

Line 35: Would you pick and choose key words that are not in the title?

Line 93: The two references are displayed as “18, 19”. Would you apply to all of the following? e.g. line 100: “21, 22”, line 110: “24, 25” and lines 118 and 119: “34, 35” in 3 page.

Line 134: Why are only the asterisks in this figure red?  I think better to indicate the number of individuals used in your experiment (n=@)?

Line 158: I also think better to indicate the number of individuals used (n=@) in this figure?

Line 186: “sl” should be italicized.

Lines 214 and 215: Figure caption should be as follow: Seasonal changes of thyroid hormones (T4 and T3) and steroid hormones (E2, T and 11‐KT) for each silvering stage of female Japanese eels from both the lake and inlets.  I also think better to indicate the number of individuals used (n=@) in this figure?

Lines 278-281: What number is Okamura et al. 2002?

Lines 325 and 330: The correct figure number is 5.

Lines 333: The correct figure number is 6.

Line 346: Which p is capital or lowercase letter in this manuscript?

Author Response

The authors presented an interesting knowledge about the silvering process in anguillid eels and the dynamics of mRNA expression of neuro hormones and pituitary hormones, thyroid hormones and sex steroids associated with the onset of their spawning migration. This manuscript has a merit to be published in Biology. There are some suggestions which would improve the quality of this manuscript.

Line 35: Would you pick and choose key words that are not in the title?

- The present title contains two keywords (Anguillid eels and Onset of spawning migration), and the other two keywords (Silvering and 11-ketotestosterone) are related to the title. Usually, the keywords should be comprehensive to cover the study, regardless of what words are in the title.

Line 93: The two references are displayed as “18, 19”. Would you apply to all of the following? e.g. line 100: “21, 22”, line 110: “24, 25” and lines 118 and 119: “34, 35” in 3 page.

- According to this comment, we corrected the writing style when two references are cited.

Line 134: Why are only the asterisks in this figure red?  I think better to indicate the number of individuals used in your experiment (n=@)?

Line 158: I also think better to indicate the number of individuals used (n=@) in this figure?

Lines 214 and 215: Figure caption should be as follow: Seasonal changes of thyroid hormones (T4 and T3) and steroid hormones (E2, T and 11‐KT) for each silvering stage of female Japanese eels from both the lake and inlets.  I also think better to indicate the number of individuals used (n=@) in this figure?

- According to these comments, we added the number of individuals in the figures. We also changed the color of asterisks in figure 2.

Line 186: “sl” should be italicized.

Lines 325 and 330: The correct figure number is 5.

Lines 333: The correct figure number is 6.

- According to these comments, we corrected above points.

Lines 278-281: What number is Okamura et al. 2002?

- We forgot to include Okamura et al. 2002 in references, so we added it.

Line 346: Which p is capital or lowercase letter in this manuscript?

- We unified the notation of p-vale to lowercase p.

Reviewer 2 Report

The authors employed anguillid eels as examples to summarize endocrinological influences on the onset of spawning migration of fishes. In general, the overall writing is ok. However, major revisions are still required before acceptance for publication.

  1. Extra editing is necessary. For example, please rewrite these sentences in lines 11-12, 18-19, 124-126, and 155-157. Lines 127: change “was” to “were”.
  2. Abstract: Add more details about other hormones except for 11-KT.
  3. Provide the full name for any abbreviated term at its first appearance in the main text (such as Prl in line 143, sl in line 144, and SE in line 162).
  4. Figures 2 to 4: Numbers of examined samples should be documented.
  5. Update your references with recent advancements. In fact, over 90% of cited documents were published at least 15 years ago.

Author Response

The authors employed anguillid eels as examples to summarize endocrinological influences on the onset of spawning migration of fishes. In general, the overall writing is ok. However, major revisions are still required before acceptance for publication.

  1. Extra editing is necessary. For example, please rewrite these sentences in lines 11-12, 18-19, 124-126, and 155-157. Lines 127: change “was” to “were”.

- In revised manuscript, English grammar and wording were checked by professional English reviewer.

  1. Abstract: Add more details about other hormones except for 11-KT.

- We agree that additional explanations regarding other hormones would be better understood by readers. However, due to the abstract word limit, we did not add more about other hormones.

  1. Provide the full name for any abbreviated term at its first appearance in the main text (such as Prl in line 143, sl in line 144, and SE in line 162).

- As for Prl and Sl, the full name was written in this paragraph. As for SE, we changed “SE” to “standard error”.

  1. Figures 2 to 4: Numbers of examined samples should be documented.

- According to this comment, the number of samples were added in the figures.

  1. Update your references with recent advancements. In fact, over 90% of cited documents were published at least 15 years ago.

- As you indicated, many of the cited references are to articles that were published more than 15 years ago. Since research in this field requires both field studies to catch the eels and physiological experiments, it is a relatively slow progressing field. In this manuscript, important articles that are published recently are covered. No other reviewers have pointed this out to us or mentioned papers we missed. Therefore, we think there is no need to add more literature.

Reviewer 3 Report

The manuscript "Anguillid eels as a model species for understanding endocrinological influences on the onset of spawning migration of fishes" appear to be an easy-to-understand description of endocrine influences on the onset of spawning migration of fish. In my opinion, it was not considered a very technical and in-depth manuscript in relation to the proposed topic. But I found it very useful and concise material on the endocrine system and its influence on the silvering and migration of fish for spawning. 
However, I propose some corrections to accept manuscript publication

1. Editing of English will be necessary for minor grammar corrections

2. According to the instructions for manuscript submission, the "Simple Summary Section" is not necessary (please check). 

3. The abstract should be a total of about 200 words maximum (please check).

4. The references were not described following the instructions

Author Response

The manuscript "Anguillid eels as a model species for understanding endocrinological influences on the onset of spawning migration of fishes" appear to be an easy-to-understand description of endocrine influences on the onset of spawning migration of fish. In my opinion, it was not considered a very technical and in-depth manuscript in relation to the proposed topic. But I found it very useful and concise material on the endocrine system and its influence on the silvering and migration of fish for spawning. 
However, I propose some corrections to accept manuscript publication.

  1. Editing of English will be necessary for minor grammar corrections.

- According to this comment, our manuscript was reviewed by a professional English editor.

  1. According to the instructions for manuscript submission, the "Simple Summary Section" is not necessary (please check). 

- Editor told us that “Simple Summary Section” is required, so we did not delete it.

  1. The abstract should be a total of about 200 words maximum (please check).

- We reduced the words in Abstract.

  1. The references were not described following the instructions

- According to this comment, we revised the references.

Round 2

Reviewer 2 Report

In fact, the authors didn't make corrections according to the reviewer's comments. I therefore provide my original opinions again for your consideration.

The authors employed anguillid eels as examples to summarize endocrinological influences on the onset of spawning migration of fishes. In general, the overall writing is ok. However, major revisions are still required before acceptance for publication.

1. Extra editing is necessary. For example, please rewrite these sentences in lines 11-12, 18-19, 124-126, and 155-157. Lines 127: change “was” to “were”.

2. Abstract: Add more details about other hormones except for 11-KT.

3. Provide the full name for any abbreviated term at its first appearance in the main text (such as Prl in line 143, sl in line 144, and SE in line 162).

4. Figures 2 to 4: Numbers of examined samples should be documented.

5. Update your references with recent advancements. In fact, over 90% of cited documents were published at least 15 years ago.

Author Response

In the revised manuscript, we have made many corrections according to reviewer’s comments. We checked the reviewer’s previous comments and all of those were fixed and are ok now except for no.2 and 5, and our responses remain the same as before. In addition, more English checking and editing has been done by an English native editor who studies fish ecology and writes many articles. Thus, we think we have sufficiently revised the manuscript according to the reviewer’s comments.